# Analysis and Measurement of Differential-Mode Magnetic Noise in Mn-Zn Soft Ferrite Shield for Ultra-Sensitive Sensors

**DOI:** 10.3390/ma15238704

**Published:** 2022-12-06

**Authors:** Danyue Ma, Xiujie Fang, Jixi Lu, Kun Wang, Bowen Sun, Yanan Gao, Xueping Xu, Bangcheng Han

**Affiliations:** 1School of Physics, Beihang University, Beijing 100191, China; 2Key Laboratory of Ultra-Weak Magnetic Field Measurement Technology, Ministry of Education, School of Instrumentation and Optoelectronic Engineering, Beihang University, Beijing 100191, China; 3Zhejiang Provincial Key Laboratory of Ultra-Weak Magnetic-Field Space and Applied Technology, Hangzhou Innovation Institute of Beihang University, Hangzhou 310000, China

**Keywords:** magnetic shield, Mn-Zn ferrite, differential-mode magnetic noise, SERF magnetometer

## Abstract

The magnetic noise generated by the ferrite magnetic shield affects the performance of ultra-sensitive atomic sensors. Differential measurement can effectively suppress the influence of common-mode (CM) magnetic noise, but the limit of suppression capability is not clear at present. In this paper, a finite element analysis model using power loss to calculate differential-mode (DM) magnetic noise under a ferrite magnetic shield is proposed. The experimental results confirm the feasibility of the model. An ultrahigh-sensitive magnetometer was built, the single channel magnetic noise measured and the differential-mode (DM) magnetic noise are 0.70 fT/Hz^1/2^ and 0.10 fT/Hz^1/2^ @30 Hz. The DM magnetic noise calculated by the proposed model is less than 5% different from the actual measured value. To effectively reduce DM magnetic noise, we analyze and optimize the structure parameters of the shield on the DM magnetic noise. When the outer diameter is fixed, the model is used to analyze the influence of the ratio of ferrite magnetic shielding thickness to outer diameter, the ratio of length to outer diameter, and the air gap between magnetic annuli on DM magnetic noise. The results show that the axial DM magnetic noise and radial DM magnetic noise reach the optimal values when the thickness to outer diameter ratio is 0.08 and 0.1. The ratio of length to outer diameter is negatively correlated with DM magnetic noise, and the air gap (0.1–1 mm) is independent of DM magnetic noise. The axial DM magnetic noise is less than that of radial DM magnetic noise. These results are useful for suppressing magnetic noise and breaking through the sensitivity of the magnetometer.

## 1. Introduction

Passive magnetic shields can attenuate the magnetic field of the environment and make their internal magnetic field close to zero, which is widely used in biological magnetic measurement [1,2,3], frontier scientific experiments [4,5], and ultra-high precision physical measurement [6,7,8].

Mu-metal materials with high permeability (>30,000) are widely used in passive magnetic shields [9,10,11]. However, the high electrically conducting mu-metal materials also generated high intrinsic magnetic noise owing to Johnson current, which becomes the main factor limiting the sensitivity of many atomic sensors, especially ultra-high sensitive atomic magnetometers [12,13,14]. MnZn ferrite materials with high resistivity (~1 Ω·m) and high permeability have been increasingly used to provide magnetic shielding performance but have much lower magnetic noise than that of mu-metal materials [15,16,17,18]. It has been gradually applied to many atomic sensors, including atomic magnetometer [1,2,3], atomic gyroscope, low field nuclear magnetic resonance, etc. Ferrite materials were first used in magnetic shields by Kornack et al. and the sensitivity of 0.75 fT/Hz^1/2^ in the ferrite shield was achieved [19]. It is 25 times lower than that of the mu-metal shield in a comparable size. Although the use of ferrite materials significantly reduces the amplitude of magnetic noise, it is still much higher than the quantum noise and probe light noise of the atomic magnetometer, which is the most critical factor limiting the further improvement of the sensitivity, and further improving the sensitivity can discover new natural phenomena and reveal new scientific laws [19]. By subtracting the signals from adjacent sensors, the CM magnetic noise is partially eliminated and the DM noise is obtained, which can improve the sensitivity. It has been used in many kinds of ultra-high sensitivity sensors [20,21,22].

Many groups have studied differential measurement [23,24]. Differential measurement has been widely used in super conductive quantum interference devices [25,26], and then used in the atomic magnetometer. The magnetic noise gradient and the distance between the measurement points are the most important factors influencing the reduction capability of the CM magnetic noise for differential measurement. After correcting the small difference of absolute sensitivity between two adjacent magnetometer channels, a first-order gradiometer was formed by Kominis et al. [12]. Lee et al. studied the magnetic noise of the magnetic shielding structure, and explained that the ability of differential measurement to reduce the magnetic noise can also be analyzed with fluctuation dissipation theory [15,27,28]. However, there are few studies on the DM magnetic noise of a ferrite magnetic shield.

In this study, by calculating the hysteresis loss of ferrite material, the theoretical analysis model of the DM magnetic noise of a ferrite shield is established by using the finite element method (FEM) with element software (ANSYS Maxwell 3D 16.0). We built a magnetometer experimental setup, measured the differential sensitivity of the device in a ferrite magnetic shield, and separated the DM magnetic noise. The calculated results are in good agreement with the experimental results, which proves the accuracy of the FEM model. Finally, according to the proposed calculation model, the influence of ferrite magnetic shielding thickness to outer diameter ratio, length to outer diameter ratio and the air gap between magnetic annuli on the DM magnetic noise is analyzed using differential measurement.

## 2. Methods

### 2.1. Magnetic Field Response and Differential Measurement Model

The ultra-sensitive magnetometer experimental setup based on atomic spin effect usually uses a two-light configuration in which the pump light and probe light is perpendicular to each other. The time evolution process of alkali atom spin dynamics can be solved analytically by using the Bloch equation [21,29,30]
(1)$$\frac{d}{dt}P=\frac{1}{Q}\left[\gamma_{e}P\times (B_{res}+B)+R_{P}\left(s\vec{z}-P\right)-R_{\mathrm{rel}}P\right]$$
where ***P*** is the electron spin polarization vector, ***B*** is the magnetic field vector to be measured, ***B***_res_ is the residual interference magnetic field, *Q* is the slowing-down factor. *γ_e_* is the gyromagnetic ratio, *R*_P_ is the pumping rate, and *s* is the average photon spin. *R*_rel_ is the spin-relaxation rate.

Spin polarization can be detected by the rotation angle *θ* of the polarization plane of the linearly polarized light passing through the polarized atom. When the probe light direction is along the *x*-direction, the rotation angle is
(2)$$\theta =\frac{\pi }{2}lnr_{e}cP_{x}^{e}[-A_{D1}\frac{f_{\mathrm{pr}}-f_{D1}}{{\left(f_{\mathrm{pr}}-f_{D1}\right)}^{2}-{\left(\Gamma_{D1}/2\right)}^{2}}+A_{D2}\frac{f_{\mathrm{pr}}-f_{D2}}{{\left(f_{\mathrm{pr}}-f_{D2}\right)}^{2}-{\left(\Gamma_{D2}/2\right)}^{2}}]$$
where *l* is the interaction distance between the probe light and the polarized atom. $\Gamma_{D1}$ and $\Gamma_{D2}$ are the pressure broadening of D1 and D2 lines. $f_{\mathrm{pr}}$ is the frequency of the probe light. *A*_*D*1_ and *A*_*D*2_ are the oscillation intensity of probe light D1 and D2 lines, respectively. $f_{D1}$ and $f_{D2}$ are the resonant frequencies of probe light D1 and D2 lines, respectively.

In order to suppress the influence of the light intensity and frequency fluctuation of the probe light on the measurement sensitivity, a photo elastic modulator (PEM) can be used to modulate the optical rotation angle at high frequency. The photoelectric detector is used to convert the optical signal into an electrical signal, and finally the output voltage of the magnetometer is
(3)$$V_{\mathrm{out}}=\eta I_{0}\alpha e^{-\mathrm{OD}(\nu )}\theta$$
where *η* is the conversion coefficient of the photodetector, *I*_0_ is the incident light intensity of the detection light, *α* is the modulation angle of the photo elastic modulator, and OD is the optical depth. When the interference magnetic field is compensated, the magnetic field to be measured is small and only in the *y*-direction, and *θ* ∝ *B*. In order to further suppress the influence of magnetic noise, the common method is to take the magnetic noise in the central area of the magnetic shield as common mode noise and suppress it by differential measurement [20]. The magnetic noise that cannot be suppressed is named DM magnetic noise. The magnetic field measurement constructed by differential measurement is shown in Figure 1. Two beams are used to detect the magnetic fields at different positions. After being received by two detectors, the voltage values of these two signals are directly subtracted by a differential circuit to eliminate common mode magnetic noise.

The reduction capability of differential measurement on CM magnetic noise can be calculated by the frequency response and power spectral density of the magnetometer [20]. In order to facilitate the analysis of the differential measurement method, the output signal in Equation (3) can be written as the property of frequency domain. The output signal *V*(*f*) of a channel can be represented by the theoretical frequency response function *K*(*f*), the magnetic field *B*(*f*), and the voltage noise *N*(*f*) independent of the magnetic field:(4)V(f)=K(f)B(f)+N(f)

In actual measurement, the frequency response function is also affected by the magnetic field gradient, light field gradient, temperature gradient, and temperature error, so it needs to be recalibrated. It is assumed that the actually calibrated frequency response function is *K*_est_(*f*), the output of the magnetometer will change,
(5)$$\beta \left(f\right)=\Delta K\left(f\right)B\left(f\right)+\frac{N\left(f\right)}{K_{\mathrm{est}}\left(f\right)}$$
where $\beta \left(f\right)=V\left(f\right)/K_{\mathrm{est}}\left(f\right)$ and $\Delta K\left(f\right)=K\left(f\right)/K_{\mathrm{est}}\left(f\right)$.

After the difference between the two channels, the power spectral density of the differential measurement is
(6)$$P_{s}(f)={\left|\frac{K_{1}(f)-K_{2}(f)}{K_{}^{\mathrm{est}}(f)}\right|}^{2}P[B_{\mathrm{far}}(f)]+P[\beta_{1}^{N}(f)]+{\left|\frac{K_{2}(f)}{K_{{}_{1}}^{\mathrm{est}}(f)}\right|}^{2}P[\beta_{2}^{N}(f)]$$
where $P^{}[B_{\mathrm{far}}(f)]$ is the power spectral density of far-field magnetic noise. *β*_1_*^N^* and *β*_2_*^N^* are the power spectral density of the output voltage affected by the DM magnetic noise of the first channel and second channel. When the frequency responses of the two channels of the gradiometer are exactly the same, the far-field magnetic noise can be completely eliminated by difference, but the DM magnetic noise cannot be suppressed by this method. In the next section, the theoretical measurement models of space magnetic noise distribution and the DM magnetic noise of a ferrite magnetic shield will be established.

### 2.2. Analysis of Differential Reduction Capability

The magnetic noise of the magnetic shield was analyzed based on the generalized Nyquist relation. According to this relation, the magnetic noise can be calculated by calculating the power loss *P* of the magnetic shield induced by the hypothetical excitation coil [19,31,32]
(7)$$\delta B=\frac{\sqrt{4kT}\sqrt{2P}}{\omega AI}$$
where *k* is the Boltzmann constant, *ω* = 2π*f* is the angular frequency, *A* is the area of the excitation coil, and *T* is the Kelvin temperature. The hypothetical excitation coil is placed in the hollow cylindrical shield and an oscillating current *I* = *I*_m_ sin(*ω*t) flows in a coil. The power loss mainly contains the eddy current loss $P_{e}=\int_{V}1/2\sigma E^{2}dv$ and the hysteresis loss $P_{h}=\int_{V}1/2\omega \mu^{\prime \prime }H^{2}dv$. *H* and *E* are the magnetic field strength and electric field intensity. $\mu^{\prime \prime }$ represents the imaginary part of the complex permeability $\mu =\mu^{\prime }-i\mu^{\prime \prime }$. *σ* is the electrical conductivity. We cannot measure the loss of the magnetic shield by placing the excitation coil in the magnetic shield by experiment and the actual impedance of the excitation coil cannot be ignored. The loss caused by the current generated by the hypothetical excitation loop is indirectly calculated by the complex permeability and conductivity of the magnetic shielding material. For metal materials, such as silicon steel sheet and mu-metal, the conductivity is high and the eddy current noise is large (*σ* ~ 10^6^ Ω^−1^ m^−1^), resulting in a large magnetic noise, about ~10 fT/Hz^1/2^ [11]. For the ferrite materials, the conductivity is very weak (*σ* ~ 10 Ω^−1^ m^−1^), so the eddy current loss is very small and hysteresis loss is dominant. Magnetic noise of the ferrite shield is less than ~1 fT/Hz^1/2^ [14]. The radial magnetic noise at the center of the magnetic shield can be rewritten as
(8)$$\delta B_{}^{T}=\frac{\sqrt{4kT}\sqrt{\int_{V}\mu^{\prime \prime }H_{}{}^{2}dV}}{\sqrt{\omega }AI}$$
where $\int_{V}H_{}{}^{2}dV$ can be numerically calculated using FEM. During FEM simulation, a single turn excitation coil is placed at the center of the hollow cylindrical shield as shown in Figure 2a, and the volume fraction of the magnetic field intensity of the magnetic shield is calculated using the finite element software (ANSYS Maxwell 3D 16.0). Finally, the magnetic noise is calculated. Ferrite materials have a low permeability and shielding coefficient, so they are usually placed inside the magnetic shield of multi-layer mu-metal to shield the noise of mu-metal, and the noise generated by itself is low. Because the residual magnetic field in the mu-metal shield is small, the complex permeability is a fixed value in the Rayleigh range of ferrite, independent of frequency, which is proved in Reference [14]. The magnetic noise produced by the magnetic shield belongs to far source magnetic noise, which can be suppressed by differential method, and the reduction capability can be simulated by FEM. One excitation coil of the single turn is replaced by the gradient coil.

The FEM simulation diagram is shown in Figure 2b. As mentioned in the previous section, the magnetic noise is still calculated by the power loss *P* of the magnetic shield caused by the excitation coil. However, in the differential method, the excitation coil is changed from a single coil to a pair of gradient coils, and the central line of the coils is called the baseline length *d* [14]. Gradient configuration is divided into axial gradient configuration and radial gradient configuration. The baseline of the gradient coil in the axial gradient configuration coincides with the axial center line, and the gradient coil in the radial gradient configuration coincides with the radial center line. The DM magnetic noise can be expressed as
(9)$$\delta B_{\mathrm{diff}}=\frac{\sqrt{4kT}\sqrt{\int_{V}\mu^{\prime \prime }H_{\mathrm{diff}}{}^{2}dV}}{\sqrt{\omega }AI}$$
where $\int_{V}H_{\mathrm{diff}}{}^{2}dV$ was numerically calculated with the gradient configuration. The reduction capability can be evaluated by the ratio of magnetic noise to DM magnetic noise $\delta B/\delta B_{\mathrm{diff}}$.

To prevent the influence of the parameters set by the simulation on the results, we verified this by changing I from 0.1 to 10 A and the diameter of loop from 1 to 20 mm in the simulation, and the simulation results were approximately the same. In this paper, the global adaptive grid is used to encrypt the grid at the location with large local error (such as the gap), while the grid at other locations is slightly thicker. By setting different grid sizes and comparing the output results, we not only improve the calculation accuracy, but also reduce the simulation time and resource allocation. The suppression effect of the gradient measurement can be further analyzed by the finite element method.

## 3. Experimental Setup and Results

### 3.1. Magnetic Field Response and Differential Measurement Model

The main components of MnZn ferrite are Fe_2_O_3_, Mn_3_O_4_, and ZnO. The doping of trace elements is used to improve the performance of ferrite, obtain higher mechanical and magnetic properties, improve permeability, and lower hysteresis loss. SiO_2_ can improve the temperature coefficient and mechanical strength of soft ferrite, but it will increase the loss. In order to reduce its adverse effects, CaO is added to improve the resistivity and reduce the loss. A small amount of doping of Co^2+^ and Ti^4+^ is also used to improve temperature characteristics and reduce losses. The XRD patterns of Mn_x_Zn_1−x_Fe_2.06_O_4_ samples are presented in Figure 3. The XRD pattern of Mn-Zn ferrite, which is in good agreement with the standard JCPDS (10-0467), indicates the sample is in the spinel phase structures. The average lattice constant a of the ferrite sample is 8.50 A. The crystallite size is estimated to be 228.6 nm based on the Scherrer’s formula, and the main peak occurs at 35°.

During the ferrite molding process, the molding pressure and temperature were optimized. The final pressure was 130 MPa and the sintering temperature was 1400 °C, and the sintering was carried out under the protection of nitrogen.

In order to accurately calculate the magnetic noise, the complex permeability of ferrite materials is also measured by the experimental impedance analysis method. The real part of the relative complex permeability $\mu_{0}^{\prime }$ is 10,500 and the imaginary part of the relative complex permeability $\mu_{0}^{\prime \prime }$ is 155.13.

### 3.2. Magnetic Noise Measurement

The loss generated by the gradient coil is proposed to simulate the differential reduction capability in the ferrite magnetic shield. To verify the accuracy of the calculation, the spin-exchange relaxation-free (SERF) magnetic field measurement device is built to measure the DM magnetic noise. A spherical Pyrex cell (25 mm in diameter) contains a drop of potassium, 50 Torr nitrogen, and 1.3 atm ^4^He. The cell is heated to 208 °C through an electric heating oven. The magnetic shielding system consists of five layers of mu-metal magnetic shield and one layer of ferrite magnetic shield. At low frequency, the shielding coefficient exceeds 10^6^. The ferrite shield is assembled with five annuli and two end caps, as shown in Figure 4. The height of each annulus is 45 mm and the thickness of the end cap is 10 mm. The outer diameter of the ferrite magnetic shield is about 140 mm and the thickness is 13 mm. The air gap between annulus is about 0.1 mm. This is because the ferrite magnetic shield is composed of multiple annuli. The ceramic adhesive is used for bonding between annuli. The thickness of the glue leads to the air gap, which can be measured with a feeler gauge. The width of the air gap can also be controlled by the thickness of the glue. The residual magnetic field inside the magnetic shield is about 0.3 nT and the gradient is about 0.5 nT/cm. The residual magnetic field and magnetic field gradient are compensated in situ by the triaxial coil. The circularly polarized pump beam and linearly polarized probe beam cross orthogonally and illuminate cells at the same time. The probe light is modulated by a PEM, and collected by two detectors at the same time—see Figure 5a. In order to prevent the influence of vibration on the measurement, all optical devices and magnetic shielding systems are placed on the mechanical vibration isolation platform, as shown in Figure 5b.

Figure 6 shows the magnetic field sensitivity of the magnetic field measuring device, including single channel sensitivity, differential sensitivity, and various noises. The 30 Hz peak in the figure is the applied 30 Hz, 10 pTrms calibration signal, which is used to calculate the sensitivity. The sensitivity (red solid line) after gradient difference is significantly less than that of single channel sensitivity (purple solid line). The probe noise (blue solid line) is close to the gradient differential sensitivity. We use differential sensitivity δ*B*_diff_ to subtract probe noise δ*B*_probe_ to extract DM magnetic noise (green dotted line). DM magnetic noise is 0.1 fT/Hz^1/2^@30 Hz. According to the comparison between probe noise and DM magnetic noise, it can be seen that in the low-frequency stage (below 30 Hz), probe noise and DM magnetic noise jointly affect the improvement of differential sensitivity, and in the high-frequency stage (above 30 Hz), probe noise is the most important factor. The black solid line is the simulation result of gradient noise using the method proposed in Section 2. The main reason for the difference between the measured noise and the theoretical calculation noise in the low-frequency stage is the influence of low-frequency vibration noise. The DM magnetic noise calculated by the proposed model is less than 5% different from the actual measured value. The calculated results are in good agreement with the actual measured results, which proves the correctness of the proposed model.

## 4. Noise Reduction Capability of Differential Measurement

After proving the effectiveness of the calculation model, in order to further increase the differential sensitivity in the future, we carry out parameter analysis and optimization to provide guidance. In the study, the real and imaginary parts of the relative complex permeability are actually measured. The resistivity of the ferrite magnetic shield is about 1 Ω·m, and the cut-off frequency is much higher than the measurement bandwidth of the magnetometer. Therefore, within 100 Hz, the imaginary part and real part of complex permeability do not change with frequency. Firstly, in order to evaluate the reduction capability, we first analyzed the magnetic noise of the ferrite magnetic shield under non differential measurement. In different application fields [14,27,32,33,34], the outer diameters of ferrite magnetic shielding are usually 60 mm, 140 mm, and 200 mm. In this paper, the above dimensions of magnetic shielding are used. When the outer diameter is fixed (*D* = 60 mm, 140 mm, 200 mm), we analyze the influence of the ratio *β* of ferrite magnetic shielding thickness to outer diameter on magnetic noise. The curve of magnetic noise of the ferrite magnetic shield with *β* is shown in Figure 7. The variation law of magnetic noise is the same as that of reference [14], which also proves the accuracy of this model. The optimal value of radial magnetic noise (a) is always *β* = 0.14, while the optimal value of axial magnetic noise (b) is *β* = 0.16, and the axial magnetic noise is less than the radial magnetic noise.

In addition, we also explored the relationship between DM magnetic noise and *β* when the outer diameter is fixed, as shown in Figure 8. Differential measurement can significantly suppress the magnetic noise and the DM magnetic noise increases with the increase of the baseline *d*. With the increase of *β*, the radial and axial DM magnetic noise increases initially and decrease afterwards, and there is an optimal value. When *β* = 0.08, the radial DM magnetic noise (a) reaches the minimum, which is less than the thickness when the radial magnetic noise reaches the minimum, *β* = 0.14. When the structural parameters of ferrite magnetic shielding are the same, the axial DM magnetic noise (b) is less than that of radial DM magnetic noise. When *β* = 0.1, the axial DM magnetic noise reaches the minimum. The results of the above analysis are applicable to ferrite magnetic shielding of any outer diameter.

Table 1 summarizes the magnetic noise and DM magnetic noise under optimal *β*. Under any outer diameter, the axial reduction capability is greater than the reduction capability, and the larger the outer diameter, the greater the difference. The outer diameter increased from 60 mm to 200 mm, and the difference of axial and radial reduction capability increased from 1.03 to 5.02. When the baseline length increased from 0.5 mm to 2 mm, the reduction capability of any outer diameter decreased by 72%. Reducing the magnetic noise of ferrite magnetic shielding can effectively reduce the DM magnetic noise and improve the reduction ability.

The ratio of length to outer diameter α of ferrite magnetic shielding is also a main parameter for designing magnetic shielding results. Reference [14] shows that when α is greater than 2.5, the magnetic noise remains unchanged with the change of α. This paper explores the relationship between DM magnetic noise and α, and further improves the design of the ferrite magnetic shielding structure. When α is greater than 3, the change rate of axial DM magnetic noise and radial DM magnetic noise with α is less than 1%, as shown in Figure 9. It is considered that the DM magnetic noise is independent of α. When α is less than 1, the axial DM magnetic noise is greater than the radial DM magnetic noise, and the influence of α on the axial DM magnetic noise is more intense. When α is greater than 1, the axial DM magnetic noise is less than the radial DM magnetic noise.

We use the same method to discuss the influence of the air gap of the magnetic annuli on the DM magnetic noise. The size of the air gap of the magnetic annuli will not affect the optimal thickness. When α is greater than 3 and consists of five magnetic annuli, the air gap (*w* = 0.1–1 mm) does not affect the DM magnetic noise at the optimal thickness.

In a word, differential measurement can significantly suppress magnetic noise and the lower baseline indicates the better suppression effect. The outside diameter, thickness, and aspect ratio of ferrite will affect the effect of differential measurement. In order to achieve low gradient DM magnetic noise, the thickness of the ferrite shield should be designed to be 0.008 times the outer diameter, and the aspect ratio should be greater than 3.

## 5. Conclusions

In this study, a numerical simulation model of the influence of ferrite magnetic shielding on the gradient differential method is established. By calculating the power loss of ferrite caused by the gradient coil, the DM magnetic noise can be calculated equivalently. In order to verify the accuracy of the calculation model, we built a magnetic field measurement device to actually test the gradient magnetic noise. The measured DM magnetic noise is about 0.1 fT/Hz^1/2^, which is in perfect agreement with the calculated value.

Using this model, the influence of structural parameters of ferrite magnetic shielding (outer diameter, thickness, aspect ratio, air gap) on gradient magnetic noise is discussed. The results show that differential measurement can significantly reduce the magnetic noise, and the lower baseline indicates the better suppression effect. When the magnetic shielding diameter increases from 60 mm to 200 mm, the magnetic noise suppression ability increases by four times. When the outer diameter is fixed, the DM magnetic noise decreases first and then increases with the increase of thickness. When the thickness is 0.08 times the outer diameter, the DM magnetic noise reaches the lowest. Under the condition of optimal thickness, the air gap between the magnetic rings does not affect the DM magnetic noise. These results are useful for spatial correlation estimation and common mode magnetic noise suppression. The above results support the design of low-noise ferrite magnetic shielding and lay a foundation for improving the differential sensitivity of magnetic field measurement.

## Figures and Tables

**Figure 1 materials-15-08704-f001:**
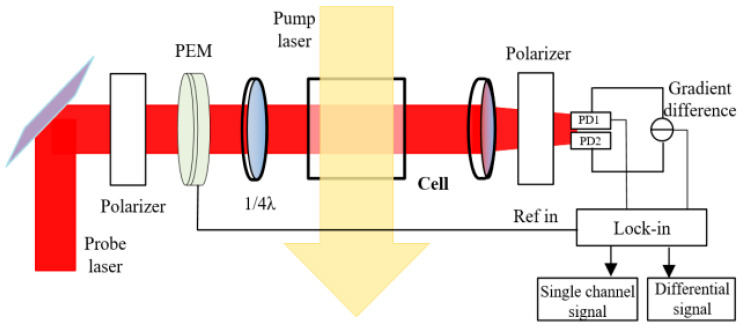
The model of differential measurement.

**Figure 2 materials-15-08704-f002:**
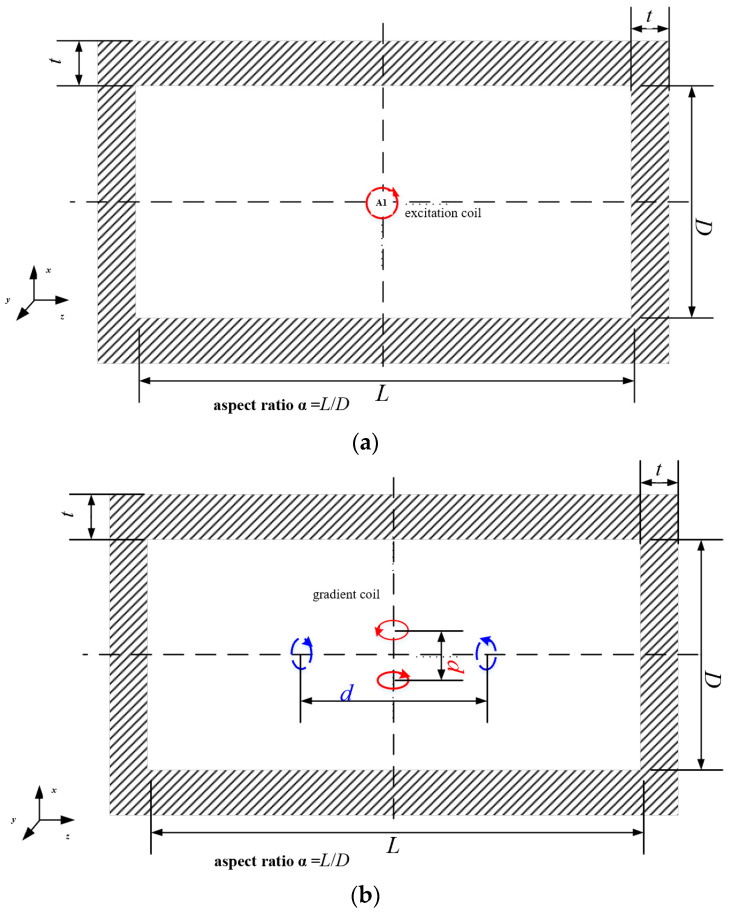
FEM simulation diagram of magnetic noise. (**a**) Simulation diagram of magnetic noise at the center of the magnetic shield, (**b**) simulation diagram of magnetic noise reduction with differential measurement configurations. The red solid line is the excitation gradient coil with axial configuration, and the blue dotted line is the excitation gradient coil with radial configuration.

**Figure 3 materials-15-08704-f003:**
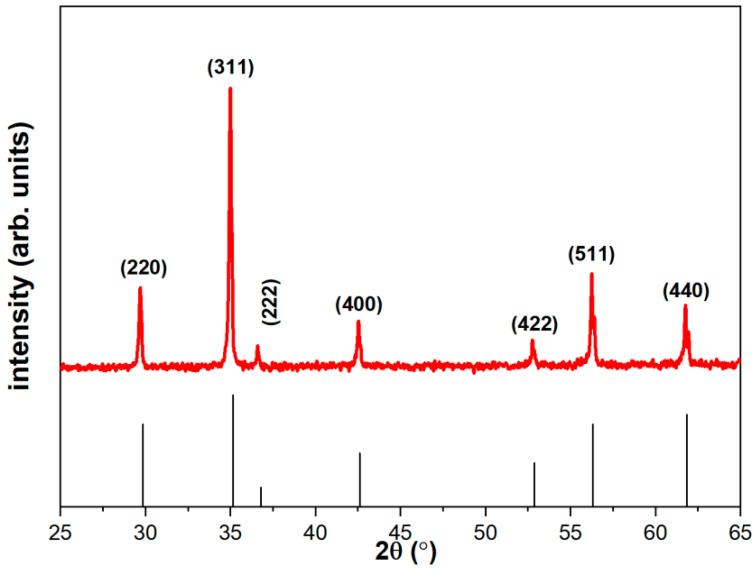
XRD patterns of Mn_x_Zn_1−x_Fe_2.06_O_4_ samples.

**Figure 4 materials-15-08704-f004:**
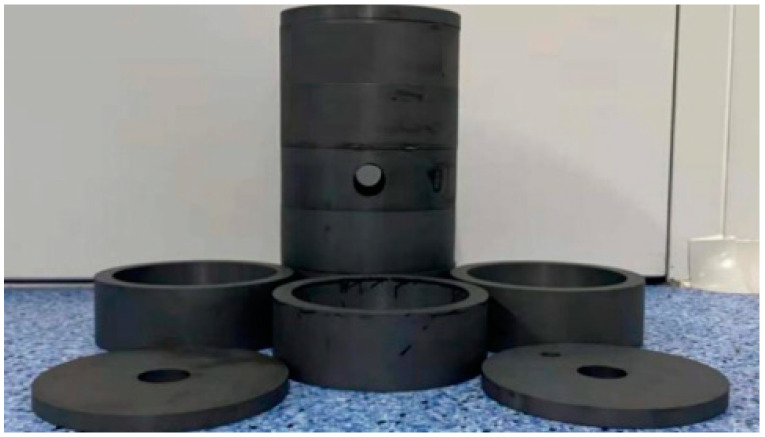
Ferrite magnetic shield and magnetic annulus. The ferrite magnetic shield has four through holes with a diameter of 25 mm in the transverse direction and two through holes with a diameter of 35 mm in the longitudinal direction.

**Figure 5 materials-15-08704-f005:**
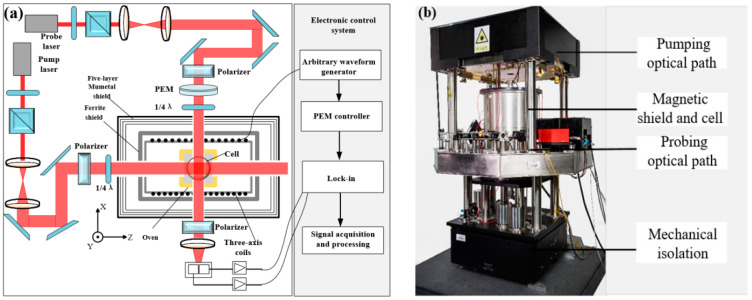
Optical path schematic diagram and actual diagram of magnetic field measuring device. (**a**) Optical path diagram under differential measurement configuration, (**b**) actual diagram of device.

**Figure 6 materials-15-08704-f006:**
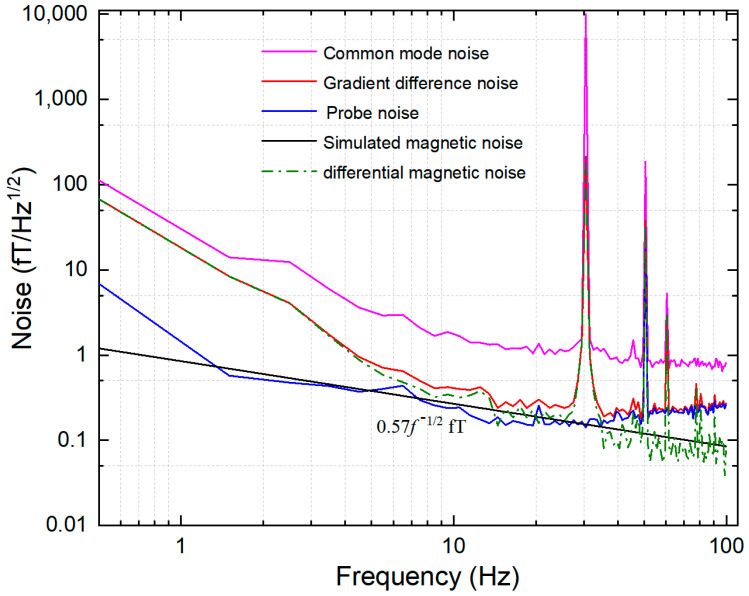
Noise spectrum of magnetic field measuring device. The purple solid line is the single channel sensitivity, about 0.7 fT/Hz^1/2^, above 30 Hz. The red solid line indicates the gradient differential sensitivity. Above 30 Hz, the differential sensitivity is about 0.18 fT/Hz^1/2^. DM magnetic noise is 0.1 fT/Hz^1/2^@30 Hz. The blue solid line indicates probe noise. The green dotted line indicates DM magnetic noise. Black is realized as simulated DM magnetic noise.

**Figure 7 materials-15-08704-f007:**
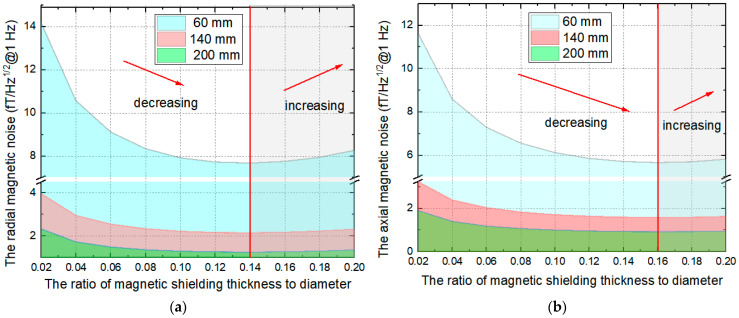
The curve of magnetic noise of ferrite magnetic shielding with *β* under different diameters, (**a**) radial magnetic noise, (**b**) axial magnetic noise.

**Figure 8 materials-15-08704-f008:**
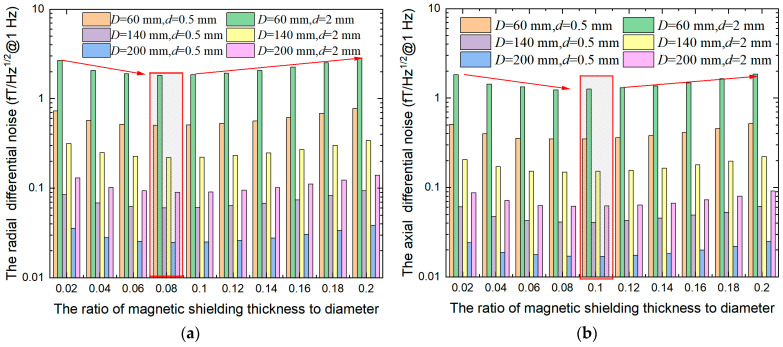
The curve of magnetic noise of ferrite magnetic shielding with *β* under different diameters and baseline length, (**a**) radial DM magnetic noise, (**b**) axial DM magnetic noise. The optimal DM magnetic noise value is in the red box.

**Figure 9 materials-15-08704-f009:**
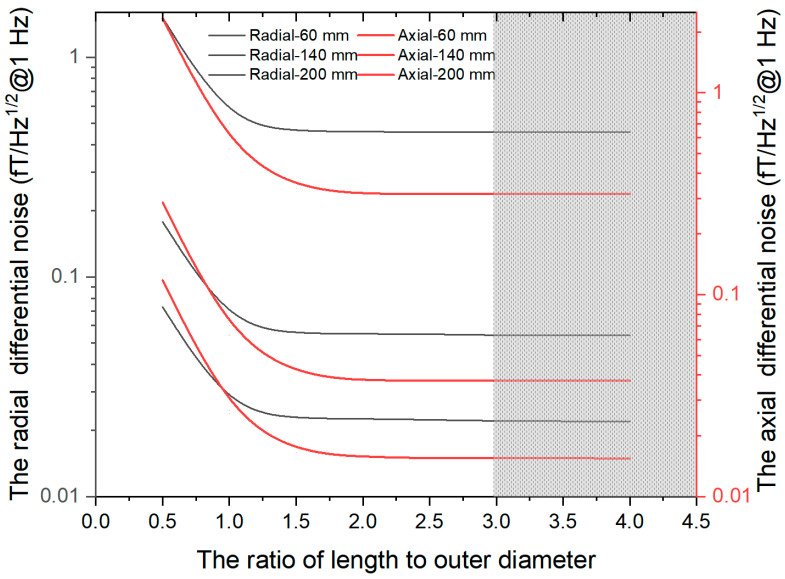
The curve of magnetic noise of ferrite magnetic shielding with α under different diameters with *d* = 0.5 mm.

**Table 1 materials-15-08704-t001:** Optimal magnetic noise and DM magnetic noise.

Magnetic Noise (fT/Hz^1/2^@1 Hz) Diameter		*D* = 60 mm	*D* = 140 mm	*D* = 200 mm
Radial magnetic noise		7.67	2.16	1.26
Axial magnetic noise		5.66	1.59	0.94
*d* = 0.5 mm	Radial DM noise	0.50	0.06	0.03
	Axial DM noise	0.35	0.04	0.02
	Radial reduction capability	15.24	35.73	50.30
	Axial reduction capability	16.27	38.89	55.32
*d* = 1.5 mm	Radial DM noise	0.96	0.12	0.07
	Axial DM noise	0.66	0.08	0.03
	Radial reduction capability	7.97	18.73	17.83
	Axial reduction capability	8.52	20.23	29.06
*d* = 2 mm	Radial DM noise	1.83	0.22	0.09
	Axial DM noise	1.23	0.15	0.06
	Radial reduction capability	4.19	9.78	13.87
	Axial reduction capability	1.35	10.49	15.11

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
