# Peer review of "Analysis and Measurement of Differential-Mode Magnetic Noise in Mn-Zn Soft Ferrite Shield for Ultra-Sensitive Sensors"

_materials, 2022, doi:10.3390/ma15238704_

Round 1

Reviewer 1 Report

The paper consists of an instrumental/experimental and computational part. An ultra-sensitive magnetometer based on atomic spin effect is constructed. It resembles the one described in the paper “High spatial resolution multi-channel optically pumped atomic magnetometer based on a spatial light modulator” by the present authors  XIUJIE FANG, KAI WEI,  TIAN ZHAO, YUEYANG ZHAI, DANYUE MA, BOZHENG XING,  YING LIU,  AND ZHISONG XIAO, published in Optics Express Vol. 28, No. 18 / 31 August 2020, 26447. The paper is not cited here. The measuring set-up is depicted in Figure 5 of the present manuscript.  A numerical algorithm and code is compiled to model the differential mode magnetic noise produced in the ferrite internal magnetic shield in addition to an external multilayer of mu-metals shields. The model is based on simulation of a pair of excitation coils that mimic the consequences of the power loss due to the fluctuation-dissipation theorem. The algorithm is argued to correctly reproduce the measured DM noise (see Figure 6) and demonstrated to be useful in designing the size and geometry of the magnetic shields. The Figures 7, 8 and 9 as well as the Table 1 illustrate the optimisation of parameters of the shielding system. The composition of the instrumental and numerical sides is a clear advantage of the work. I would suggest some amendments in presentation. In particular, in the eq. (2) the letter “f” is used for an amplitude, and the letter “nu” for frequency, whereas in eqs. (4) – (7) it is a frequency that is denoted with the letter “f”. The description of the eq. (2), essential for understanding, should be improved by stating that the probe light contains two lines of given widths, whereas the symbol nu_pr is a current variable, which is hardly legible from the phrase “ Nu_pr is the frequency of the probe light. fD1 and fD2 are the oscillation intensity of probe light D1 and D2 lines respectively. and are the resonant frequencies of the probe light D1 and D2 lines respectively.” 

Author Response

We would first like to thank the reviewer for carefully reading our paper and giving us the opportunity to improve its quality. The revisions have been highlighted in the manuscript according to the requirement. Synchronous modifications and explanations have been carried out in the revised manuscript. The specific replies and modifications are in the attached word file.

Reviewer 2 Report

The reviewer understood that the paper aimed to develop a magnetic shield for high-sensitivity magnetic field sensors. The important point of this paper, as far as the reviewer understand from the Introduction, is the analysis to suppress the noise level by using ferrite, but the explanation of the analysis method is too poor.

Although the authors said that FEM method was used, the procedure was not explained at all. Also, although a picture of the shield structure is shown, the detailed positioning in the experimental system is not described, and it is not clear at which position the noise attenuation is simulated.

The authors also mentioned about gaps, but there is no explanation of where and how gaps are provided, etc.

In general, high magnetic permeability should be related to shielding performance, but the merits of using ferrite are not clear. In fact, mu-metal or permalloy is mostly used for highly sensitive shielding. The authors described about Jonson current, but how does it works to make worse of shield performance.?

Also, the authors use complex permeability in their analysis, but this is dependent on frequency, and at frequencies as low as 100 Hz the imaginary part should normally be negligible.  What frequency was used for evaluating complex frequency? How is the effect of frequency incorporated into the analytical model? As mentioned earlier, the details of the noise analysis model are not mentioned, so the reviewer cannot give positive judge to the paper.

Generally, discontinuous domain movement largely affects the shield performance, how do the authors treat with this problem in analysis?

Author Response

(The authors gave the same response as above.)

Reviewer 3 Report

Check the corrections in pg 2/13 - explains of the reduction capability of differential measurement on magnetic noise [14].

In Pg 3 of 13, After being received by two detectors, the two signals are made difference to eliminate the common mode magnetic noise. The meaning is not conveyed properly.

In Pg 11 of 13, The results show that differential measurement can significantly reduce the magnetic noise, and the shorter the baseline, the stronger the suppression effect, English language style can be improved.

Author Response

(The authors gave the same response as above.)
